# The Impact of Variation in the Toll-like Receptor 3 Gene on Epizootic Hemorrhagic Disease in Illinois Wild White-Tailed Deer (*Odocoileus virginianus*)

**DOI:** 10.3390/genes14020426

**Published:** 2023-02-08

**Authors:** Jacob E. Wessels, Yasuko Ishida, Nelda A. Rivera, Spencer L. Stirewalt, William M. Brown, Jan E. Novakofski, Alfred L. Roca, Nohra E. Mateus-Pinilla

**Affiliations:** 1Department of Animal Sciences, University of Illinois at Urbana-Champaign, Urbana, IL 61801, USA; 2Illinois Natural History Survey—Prairie Research Institute, University of Illinois at Urbana-Champaign, Champaign, IL 61820, USA; 3Department of Pathobiology, College of Veterinary Medicine, University of Illinois at Urbana-Champaign, Urbana, IL 61802, USA

**Keywords:** bluetongue, BTV, EHDV, hemorrhagic disease, innate immunity, TLRs, TLR3, wildlife diseases, wild ungulates

## Abstract

Epizootic hemorrhagic disease (EHD) leads to high mortality in white-tailed deer (*Odocoileus virginianus*) and is caused by a double-stranded RNA (dsRNA) virus. Toll-like receptor 3 (TLR3) plays a role in host immune detection and response to dsRNA viruses. We, therefore, examined the role of genetic variation within the *TLR3* gene in EHD among 84 Illinois wild white-tailed deer (26 EHD-positive deer and 58 EHD-negative controls). The entire coding region of the *TLR3* gene was sequenced: 2715 base pairs encoding 904 amino acids. We identified 85 haplotypes with 77 single nucleotide polymorphisms (SNPs), of which 45 were synonymous mutations and 32 were non-synonymous. Two non-synonymous SNPs differed significantly in frequency between EHD-positive and EHD-negative deer. In the EHD-positive deer, phenylalanine was relatively less likely to be encoded at codon positions 59 and 116, whereas leucine and serine (respectively) were detected less frequently in EHD-negative deer. Both amino acid substitutions were predicted to impact protein structure or function. Understanding associations between *TLR3* polymorphisms and EHD provides insights into the role of host genetics in outbreaks of EHD in deer, which may allow wildlife agencies to better understand the severity of outbreaks.

## 1. Introduction

Epizootic hemorrhagic disease (EHD) is of concern to hunters, wildlife managers, and state agencies because this viral disease leads to high mortality in cervids, especially in white-tailed deer (*Odocoileus virginianus*) [1]. The etiologic agent, EHD virus (EHDV), is a double-stranded RNA (dsRNA) virus and is transmitted by biting midges of the genus *Culicoides* [2,3]. Biting midges are most active in warm weather and require specific environmental conditions to emerge after overwintering. For example, *Culicoides* do not begin to show activity until a threshold temperature of 10.9 °C (51.6 °F) is reached [4]. Furthermore, before midge emergence, the temperature cannot fall below −0.3 °C (31.5 °F) for at least seven days [4]. These conditions are rarely met in Illinois between November and March, so these months are unfavorable for vector activity or EHD transmission. Consequently, EHD outbreaks in Illinois have primarily been reported in the late summer months [5].

Seven serotypes of EHD exist worldwide [6,7], and three of these seven serotypes (EHDV-1, -2, and -6) are endemic to the United States [8,9]. In North America, EHDV affects white-tailed deer, mule deer (*Odocoileus hemionus*), pronghorns (*Antilocapra americana*), mountain goats (*Oreamnos americanus*), and other wild ungulates [2,3,9,10]. Infection with EHDV also occurs among domesticated ruminant livestock, but signs of the infection are usually minor or completely undetectable [11]. In contrast, naive (previously unexposed) white-tailed deer infected with EHDV often die within 4 to 10 days of infection [2,3,9].

The host immune system may recognize the virus upon infection and elicit an immune response. In mammals, innate and adaptive immune responses to infectious microorganisms have evolved to protect them from pathogens [12]. Toll-like receptors (TLRs), part of the innate immune system, are pattern recognition receptors [13]. TLRs are transmembrane proteins with extracellular and/or intracellular domains [14]. They recognize pathogenic microorganisms and induce signaling cascades for proinflammatory gene expression [15]. The recognition of pathogen-associated molecular patterns (PAMPs) by TLRs leads to a cascade of events that include: the upregulation of proinflammatory mediators (e.g., chemokines and cytokines), activation of the complement system, recruitment of phagocytic cells, and mobilization of professional antigen-presenting cells [15,16,17,18]. Different TLRs protect against various pathogens (e.g., viruses, bacteria, fungi, protozoa) based on their binding capacity and the specificity of the ligand to the lipid, protein, or nucleic acid components of a pathogen [12,18].

Toll-like receptor 3 is an intracellular receptor located in the endosome [14]. In mammals, TLR3 recognizes PAMPs in dsRNA viruses and elicits a cascade of immune responses [18,19]. One study examined the expression of *TLR3* mRNA in various tissues in white-tailed deer, stating that tissues with higher baseline *TLR3* expression are from organs said to be typically involved in hemorrhagic disease in deer, such as the spleen, skin, and heart [20]. A survey of DNA sequences in cattle from the *TLR3* gene detected a signature of balancing selection, leading the authors to suggest that TLR3 variability may impact differential susceptibility to dsRNA viruses [21].

We, therefore, hypothesized that the outcome of exposure of white-tailed deer to EHDV dsRNA might vary depending on the nonsynonymous *TLR3* variants present in the host. While part of the white-tailed deer *TLR3* mRNA (209 nt) has been sequenced [20], the entire coding region of the *TLR3* gene has not, to the best of our knowledge, been sequenced, nor has variation in *TLR3* been examined for associations with EHD outcomes. Here, we determined genetic variation across the entire coding region of the *TLR3* gene and examined its association with EHD in wild (not captive or fenced-in) white-tailed deer from Illinois. The goals of this study were to (1) develop primers to amplify the entire coding region of *TLR3* in white-tailed deer, (2) estimate genetic diversity within *TLR3* in Illinois wild white-tailed deer, (3) examine whether nonsynonymous SNP substitutions in *TLR3* are associated with EHD cases in Illinois white-tailed deer, and (4) examine the impacts of those SNPs on the structure and function of the TLR3 protein.

## 2. Materials and Methods

### 2.1. Study Area and Sample Collection

For this study, 84 samples from wild white-tailed deer—26 EHD-positive and 58 EHD-negative—were used. The deer spanned 21 counties across Illinois and were sampled between 2018 and 2021 by the Illinois Department of Natural Resources (IDNR). The public notifies IDNR biologists of dead deer suspected to have succumbed to EHD and of suspected outbreaks of EHD. Outbreaks of EHD in Illinois typically occur in late summer and early fall when *Culicoides* populations and activity are at their highest [2,5]. IDNR biologists respond to these reports by visiting sites with reported cases and collecting diagnostic tissues (spleen or lung) from animals found dead. Fresh tissue samples from deer with evidence of hemorrhagic disease were collected; e.g., from deer found dead near the water with a swollen tongue, face, and neck [3]. The geographic location for each animal was recorded. All tissue samples were sent to the University of Illinois Veterinary Diagnostic Laboratory (UIUC-VDL) for EHD diagnostic testing using RT-qPCR. Twenty-six deer samples that tested positive for EHD at UIUC-VDL were sequenced for this study, along with 58 control deer.

Samples for the negative control group came from white-tailed deer collected following hunter harvest, roadkill, or IDNR chronic wasting disease surveillance or management efforts. The control deer were presumed negative as they did not have signs indicative of EHD and, importantly, were collected only at a time of the year (November–March) when no active EHD outbreaks occurred in Illinois. Control deer were chosen from the same geographic locations (Figure 1) and years as the positive deer to minimize confounding factors. Due to the limited available samples, the negative controls were not always matched for age and sex; allelic frequencies should not have varied due to these characteristics. DNA was obtained primarily from spleen samples; however, in some instances, lung or skeletal muscle was used to obtain DNA.

### 2.2. Exon Determination and Primer Design

We noted that exons had been incorrectly predicted for the *TLR3* gene in the white-tailed deer genome (accession number: NW_018331484.1) available through the National Center for Biotechnology Information (NCBI). This was inferred after alignment of the exon sequences of *TLR3* for cattle (*Bos taurus*) (NM_001008664.1) and humans (*Homo sapiens*) (NM_003265.3) from the UCSC Genome Browser [22] with the predicted *TLR3* gene for white-tailed deer (NW_018331484.1). In addition, we translated three transcript variants of the predicted *TLR3* mRNA of white-tailed deer in NCBI (XM_020889930.1; XM_020889931.1; XM_020889932.1), aligning them with the amino acid sequences of humans (NM_003265) and cattle (NM_001008664), revealing that translated amino acid sequences for the white-tailed deer TLR3 were not similar to those of humans and cattle. The exon sequences were translated using the Translate tool in Expasy [23] and aligned using Clustal Omega [24]. We inferred the exon/intron boundaries, exon sequences, and coding regions by comparing the deer sequence to the exon sequences of humans (NM_003265.3) and cattle (NM_001008664.1). Together, these analyses showed that the exons of the white-tailed deer *TLR3* gene had been wrongly predicted. Our analyses established that, as is the case in humans and cattle, the white-tailed deer *TLR3* is comprised of five exons, with a coding region in four of the exons (2, 3, 4, and 5).

The *TLR3* genes of the cattle genome (GCF_002263795.1) and white-tailed deer (NW_018331484.1) were aligned and used to design primers based on conserved regions. As exon 4 is 1850 bp long, two primer pairs were used to amplify overlapping regions of exon 4. The sequences of exon 5, its flanking intron, and the 3′ untranslated region (UTR) were unavailable for the white-tailed deer; thus, the published sequence for cattle *TLR3* was used exclusively to design a primer flanking exon 5. All primers were designed using Primer 3 software [25], targeting introns or UTRs to include the complete coding regions within the amplicons. In addition to primer pairs designed for PCR, 14 internal primers were designed for Sanger sequencing, targeting exonic regions conserved between cattle and white-tailed deer. The name, sequence, target site, and purpose of each primer are listed in Table 1.

### 2.3. DNA Extraction, PCR, and Sanger Sequencing of the TLR3 Gene

DNA was extracted from tissue samples using a Qiagen DNeasy Blood and Tissue kit (QIAGEN, Valencia, CA, USA) following the manufacturer’s protocol, with a minor modification of overnight incubation for the cell lysis step to maximize DNA yield per extraction.

For exons 2, 3, 4, and 5, the PCR mix consisted of 20 ng of template DNA, 0.4 µM final concentration of each oligonucleotide primer, 1.5 mM MgCl_2_, 200 µM of each of the dNTPs (Applied Biosystems; ABI), and 1X PCR Buffer II (ABI) with 0.08 units/µL final concentration of AmpliTaq Gold DNA Polymerase (ABI). For some samples for exon 3, primer final concentration was reduced to 0.2 µM to reduce primer dimer formation, while all other reagent amounts were kept the same. Exons 2, 3, and 5 were amplified in a 25 µL reaction volume, while exon 4 was amplified in a 50 µL reaction volume since the larger volume allowed for a larger number of subsequent Sanger sequencing reactions.

PCR for exons 2 and 5 involved an initial denaturation step of 95 °C for 9.45 min. This was followed by cycles of three steps. The first step was 20 s denaturing at 94 °C. The second step was 30 s annealing at 60 °C (in the first 3 cycles); 58, 56, 54, and 52 °C (5 cycles for each temperature); and 50 °C (last 22 cycles). The third step was a 1.5 min extension at 72 °C. After the last cycle, there was a final extension step of 7 min at 72 °C (Appendix A). PCR for exon 3 involved an initial denaturation step of 95 °C for 9.45 min. This was followed by 45 cycles of three steps each: (1) 20 s denaturing at 94 °C; (2) 30 s annealing at 55 °C; and (3) 45 s extension at 72 °C. There was a final extension after the last cycle of 7 min at 72 °C. PCR for the two amplicons of exon 4 involved an initial denaturation step of 95 °C for 9.45 min. This was followed by cycles of three steps. The first step was 20 s denaturing at 94 °C. The second step was 30 s annealing at 60 °C (in the first 3 cycles); 58, 56, 54, and 52 °C (5 cycles for each temperature); and 50 °C (last 22 cycles). The third step was a 3 min extension at 72 °C. After the last cycle, there was a final extension step of 7 min at 72 °C (Appendix A).

After PCR, successful amplification was confirmed using a 1% agarose gel with ethidium bromide viewed under UV light. The dNTPs and unincorporated primers were then removed from the PCR product using Exonuclease I (New England Biolabs) and shrimp alkaline phosphatase (New England Biolabs), respectively [26].

Sanger sequencing reactions of 10 µL volume used a BigDye Terminator v3.1 Cycle Sequencing Kit (ABI). Reaction included 0.25 µL of BigDye, 1.875 µL 5X buffer, 2.5 µL of purified PCR product, and a final concentration of 0.12 µM primer. Sequences were resolved on an ABI 3730XL capillary at the Keck Center for Functional and Comparative Genomics at the University of Illinois at Urbana-Champaign. The software Sequencher version 5.4 (Gene Codes Corporation, Ann Arbor, MI, USA) was used to view and edit chromatograms, assemble contigs, and trim final sequences to include only the coding regions of our inferred *TLR3* exon sequences. All exon sequences were concatenated to include the entire coding region of *TLR3* for each deer; these were translated to establish them as in-frame. Alignment of the translated deer amino acid sequences to those of humans (NM_003265) and cattle (NM_001008664) verified that the complete coding region had been generated (Appendix A).

### 2.4. Single Nucleotide Polymorphism (SNP) Analyses

The haplotype phase of the complete sequences was inferred using PHASE [27]. PHASE utilizes a coalescent-based Bayesian method and was implemented in DnaSP version 5.10.1 [28,29] with 10,000 iterations and 100 burn-in iterations using our available sequences (*n* = 84 deer). DnaSP [28] was then used to translate each inferred haplotype into a protein sequence. MEGA 10.2.4 was used to align haplotypes and their translated amino acid sequences [30]. A median-joining network was generated for the phased data using PopART [31].

Haplotype diversity (*H_d_*), nucleotide diversity (*π*), and linkage disequilibrium (LD) were calculated using DnaSP [28]. To determine whether there was enough power to test for a potential association between a nonsynonymous SNP and the EHD test results, a power analysis was conducted using OSSE “http://osse.bii.a-star.edu.sg (accessed on 6 July 2021)”. The nonsynonymous SNP sites for which sample sizes were insufficient to achieve at least 80% power were excluded from further consideration [32]. To test whether there was an association between nonsynonymous SNPs and EHD, Fisher’s exact test was conducted using R version 3.4.0 [33]. Since our data did not involve a normal distribution—and also due to the small sample counts for some SNPs—we conducted Fisher’s exact test. The *p*-values were adjusted for multiple hypothesis testing using the Benjamini–Hochberg procedure [34] in R. For the nonsynonymous mutations demonstrating significant associations with EHD, their impact on TLR3 protein function or structure was examined using Provean version 1.1.3 [35,36] and PolyPhen-2 [37].

## 3. Results

We inferred that the white-tailed deer *TLR3* gene is comprised of five exons, as is the case for cattle and humans, with coding regions in four of the exons (exons 2, 3, 4, and 5). The translated amino acid sequences of our white-tailed deer *TLR3* sequences were homologous to those of humans (NM_003265) and cattle (NM_001008664) (Appendix A), which further confirmed that we amplified the four *TLR3* exons containing coding regions.

Using novel primers (Table 1), the entire coding region of *TLR3* (2715 base pairs encoding 904 amino acids) was successfully sequenced in 84 white-tailed deer from Illinois. Alignment of deer sequences revealed SNPs at 77 positions in the coding region (Appendix A). We detected 11 SNPs in exon 2, 57 in exon 4, and 9 in exon 5, while no SNPs were detected in exon 3 (Appendix A). Of these 77 SNPs, 45 were synonymous, and 32 were nonsynonymous. A total of 58 of the 77 SNPs were in the ligand-binding region of *TLR3*, including 26 of the nonsynonymous SNPs. Nucleotide diversity in the coding region was *π* = 0.004. There were no SNPs in the exon/intron boundary regions of the *TLR3* gene.

After sequences were phased, 85 distinct haplotypes were identified across the deer samples. These were numbered as “Haplotype 1” through “Haplotype 85” in descending order of frequency (GenBank accession numbers: OL744113–OL744197; Appendix A). Of the 85 haplotypes, only 12 had a frequency of 0.02 or higher (Appendix A). Most were at low frequencies, as illustrated by the median-joining network in Figure 2. Thus, haplotype diversity was very high (*Hd* = 0.983).

When translated, the 85 haplotypes encoded 55 different protein variants of TLR3. As most protein variants were encoded by a single haplotype, each protein variant was assigned a number corresponding to the number given to the haplotype; if more than one haplotype encoded for the same protein variant, the haplotype number with the highest frequency was used to designate the protein variant. For example, the most common protein variant, with a frequency of 0.131 (Appendix A), was encoded by Haplotype 1 and, hence, was designated TLR3-01.

Power analyses indicated that 5 of the 32 nonsynonymous SNPs within *TLR3* would represent a sufficient sample size for statistical analyses. For these five SNPs (nucleotide positions 175, 347, 718, 995, and 1007 in *TLR3;* respectively, codons 59, 116, 240, 332, and 336), we tested the association between each SNP and EHD using Fisher’s exact test adjusted for multiple hypotheses testing. Two nonsynonymous SNPs in *TLR3*, at codons 59 and 116, were significantly associated with EHD occurrence in Illinois white-tailed deer, even after Benjamini–Hochberg correction for multiple hypothesis testing (Appendix A). In EHD-negative deer, when compared to EHD-positive deer, phenylalanine (F) (1 positive and 17 negative deer) was encoded more frequently than leucine (L) (51 positive and 99 negative deer) at codon 59 (odds ratio (*OR*) = 8.69, *p* = 0.0340, Fisher’s exact test, two-tailed). In EHD-negative deer, when compared to EHD-positive deer, phenylalanine (F) (1 positive and 18 negative deer) was encoded more frequently than serine (S) (51 positive and 98 negative deer) at codon 116 (*OR* = 9.29, *p* = 0.0403, Fisher’s exact test, two-tailed). Each of these variants encoded an amino acid located in a leucine-rich repeat (LRR). The amino acid encoded by codon 59 was located in LRR1, while the amino acid encoded by codon 116 was in LRR3, based on the positions of the homologous codons for humans in UniProt “https://www.uniprot.org/uniprotkb/O15455/entry (accessed on 6 July 2021)” [38,39].

For the deer, we also determined the amino acids encoded by the two chromosomes present in each diploid individual (Figure 3). When both copies of the *TLR3* gene in an individual deer encoded L at codon 59, the deer were designated “LL”. When one copy encoded L and the other F, the deer were designated “LF”. In EHD-positive deer, LL was detected more frequently than LF relative to their frequencies in EHD-negative deer (*OR* = 10.16, *p* = 0.0088, Fisher’s exact test, two-tailed). Among EHD-positive deer, there were 25 LL and 1 LF individuals; among EHD-negative deer, there were 41 LL and 17 LF individuals (Figure 3A). For codon 59, no “FF” deer were found, consistent with the minor allele being at a low frequency (0.107 for the SNP encoding F; thus, an expected value of 0.011 for the FF genotype).

When both copies of the *TLR3* gene in an individual deer encoded S at codon 116, the deer were designated “SS”, whereas, when one copy encoded S and the other F, the deer were designated “SF”. In EHD-positive deer, SS was detected more frequently than SF compared to their frequencies in EHD-negative deer (*OR* = 11.02, *p* = 0.0048, Fisher’s exact test, two-tailed). Among EHD-positive deer, there were 25 SS individuals and 1 SF; among EHD-negative deer, there were 40 SS and 18 SF individuals (Figure 3B). For codon 116, no “FF” deer were found, consistent with the minor allele being at low frequency (0.113 for the SNP encoding F; thus, an expected value of 0.013 for the FF genotype) (Appendix A).

We also determined the combination of codons 59 and 116 for each deer. In this case, deer were designated using four letters, with the first two letters indicating the amino acids encoded by codon 59 and the last two letters indicating the amino acids encoded by codon 116. For example, deer in which both copies of *TLR3* encoded L at codon 59 and both copies encoded S at codon 116 would be designated “LLSS”. The combination LLSS was detected relatively more frequently than LFSF in EHD-positive deer relative to EHD-negative deer (*OR* = 9.67, *p* = 0.0151, Fisher’s exact test, two-tailed). Among EHD-positive deer, there were 25 LLSS individuals and 1 LFSF; in EHD-negative deer, there were 38 LLSS and 15 LFSF individuals (Figure 3C). The other combinations were very rare, so association analyses were not feasible. No positive deer were LFSS or LLSF, and only a small number of negative deer carried either of these combinations (Figure 3C). Neither EHD-positive nor EHD-negative deer carried the combination FFFF (Figure 3C).

The SNPs at codon 59 and codon 116 proved to be in linkage disequilibrium (LD) (*D* = 0.083, *p* < 0.0001). Leucine at codon 59 was detected more frequently with serine (*n* = 147) than with phenylalanine at codon 116 (*n* = 3), while phenylalanine at codon 59 was detected more frequently than expected with phenylalanine (*n* = 16) than with serine (*n* = 2) at codon 116 (Figure 2). Since these two nonsynonymous SNPs were in LD, it is possible that only one of them may have an impact on EHD. We examined the potential effect of these amino acid substitutions on protein structure and function using the software Provean [35,36] and PolyPhen-2 [37]. Both sites were predicted to have an impact on the biological function of the protein by Provean [35,36]. The software PolyPhen-2 [37] predicted that both amino acid changes would impact the structure and function of the protein, with a HumDiv score of 0.989 for codon 59 and a score of 0.999 for codon 116, both suggesting a substantial effect on the protein.

## 4. Discussion

Genetic variation in Toll-like receptors influences the frequency and outcomes of infectious diseases [40]. Studies of variation and/or selection in the *TLR3* gene have furthered our understanding of diseases in wildlife populations [41,42,43,44]. In white-tailed deer, the expression of *TLR3* mRNA is reported to be higher in tissues said to be typically involved in hemorrhagic disease [20]. In the current study, we identified the exons for *TLR3* in white-tailed deer, developed primers to evaluate the genetic diversity of *TLR3* in Illinois white-tailed deer, and evaluated the frequencies of nonsynonymous substitutions within the *TLR3* gene in EHD-positive and EHD-negative deer. To the best of our knowledge, this is the first study to sequence the complete coding region of *TLR3* in wild white-tailed deer and to analyze *TLR3* polymorphisms for associations with EHD.

We found 77 SNPs (Appendix A) and high haplotype diversity (*Hd* = 0.983; Figure 2) in the coding region of *TLR3* in wild white-tailed deer. This contrasts with the low *TLR3* diversity reported for roe deer (*Capreolus capreolus*). In a previous study that sequenced 1601 bp (part of exon 4) in 32 roe deer, only two SNPs, both synonymous, were detected, forming three haplotypes [44]. The authors hypothesized that the low diversity in roe deer could be due to purifying selection, potentially mediated by a pathogen [44]. In our white-tailed deer, the same region of *TLR3* (nucleotide positions 851 to 1576 in Appendix A) included 31 SNPs, of which 14 were nonsynonymous. The high diversity for *TLR3* in white-tailed deer is consistent with other previous studies reporting high haplotype diversity in *TLR3* in wild rabbits [43] and domesticated cattle [21,45]. For cattle, Chen et al. (2020) reported 43 *TLR3* haplotypes in 144 cattle, whereas we identified 85 haplotypes in 84 wild white-tailed deer in Illinois (Appendix A). The diversity of *TLR3* in cattle is higher than that of most other cattle Toll-like receptor genes; the high cattle *TLR3* diversity has been attributed to balancing selection [21].

We could not evaluate how pathogens, such as EHDV and BTV, may have impacted the diversity of *TLR3* in white-tailed deer. However, in cattle, exposure of populations to multiple pathogens has been hypothesized to explain signatures of balancing selection detected for *TLR3* [21]. Balancing selection can be due to heterozygote advantage, but this could not be examined in our white-tailed deer dataset because there was a dearth of homozygotes for the minor allele at the two *TLR3* nonsynonymous SNPs (codons 59 and 116) found to be associated with EHD (Figure 3). Nevertheless, the high haplotype diversity found in white-tailed deer may suggest that pathogen-mediated balancing selection could, in part, be maintaining the high *TLR3* diversity.

The nonsynonymous mutations associated with EHDV infection in white-tailed deer code for amino acids in leucine-rich repeats 1 and 3, which provide binding sites for dsRNA [38,39]. As these are involved in recognizing PAMPs, the nonsynonymous mutations shown to be associated with EHD can be hypothesized to affect the recognition of PAMPs in EHDV and/or to modulate the risk for severe infection. Examining the geographic distribution of *TLR3* SNPs among deer populations may help predict the impact of EHD outbreaks in wild white-tailed deer in different geographic regions or local areas. For example, geographic surveys of SNPs in *TLR3* may reveal herds or regions with high or low frequencies of vulnerable and less vulnerable deer. This information could be used to model and predict the severity of the outbreaks of EHDV or BTV in wild and captive deer herds.

EHD has been observed in the southern USA for over a century [3,9]. There is higher mortality at northern than southern latitudes, suggesting a more recent exposure of immunologically naive white-tailed deer in northern populations to EHDV [2,9] or cross-protection from previous exposure to different EHDV serotypes in southern populations [46]. In Illinois, multiple EHD outbreaks have been reported across the state during the past few decades, with epizootic and enzootic levels of reported cases in white-tailed deer populations across the state [5]. Our samples were from 22 counties, representing deer populations in south, central, and northern Illinois (Figure 1). The Illinois land used by wild white-tailed deer is predominantly agricultural, with mixtures of prairies, wetlands, and forests, and may include areas with moderate to high urban densities [47,48]. Mixed agricultural landscapes have been reported to favor exposure to a diverse number of pathogens [49]. In these areas, pathogen exposure is highly heterogenous, which may favor the maintenance of high allele diversity at *TLR* genes [44].

Various studies have linked infectious diseases in animals and humans with *TLR* gene polymorphisms [50,51,52]. However, TLR diversity in wildlife species and its association with habitat disturbance and diverse pathogen and vector–host interactions have been less commonly investigated [53]. Studies of the role of host–vector–pathogen interactions and the effect of multiple pathogen infections in natural environments would advance our understanding of the role of these factors in EHD [50]. In other Toll-like receptor genes, landscape characteristics and exposure to pathogens are believed to affect allelic diversity [21,50,52,54]. Landscape characteristics and temporal changes in vector habitat and pathogen exposure could, thus, play a role in the high diversity of *TLR3* in wild white-tailed deer. Future studies could explore the effects of ecological factors (e.g., habitat, host population density) and anthropogenic changes in relation to immunogenetics and wildlife health [52].

## 5. Conclusions

We successfully sequenced the complete coding region of *TLR3* (2715 base pairs encoding 904 amino acids) in 84 Illinois wild white-tailed deer and demonstrated that *TLR3* haplotype diversity is high in this species. We identified nonsynonymous substitutions at codons 59 and 116 in the *TLR3* gene that showed significantly different frequencies between EHD-negative and EHD-positive deer, suggesting that these substitutions may influence vulnerability to EHD. Codons 59 and 116 were in linkage disequilibrium, and both substitutions were predicted to impact TLR3 structure and function; thus, it is possible that either or both codons may influence EHD in white-tailed deer. Combining the role of TLR3 on innate immunity against EHDV with surveillance of vectors, circulating viruses, and vector habitat characteristics may help predict the severity of outbreaks of epizootic hemorrhagic disease among white-tailed deer.

## Figures and Tables

**Figure 1 genes-14-00426-f001:**
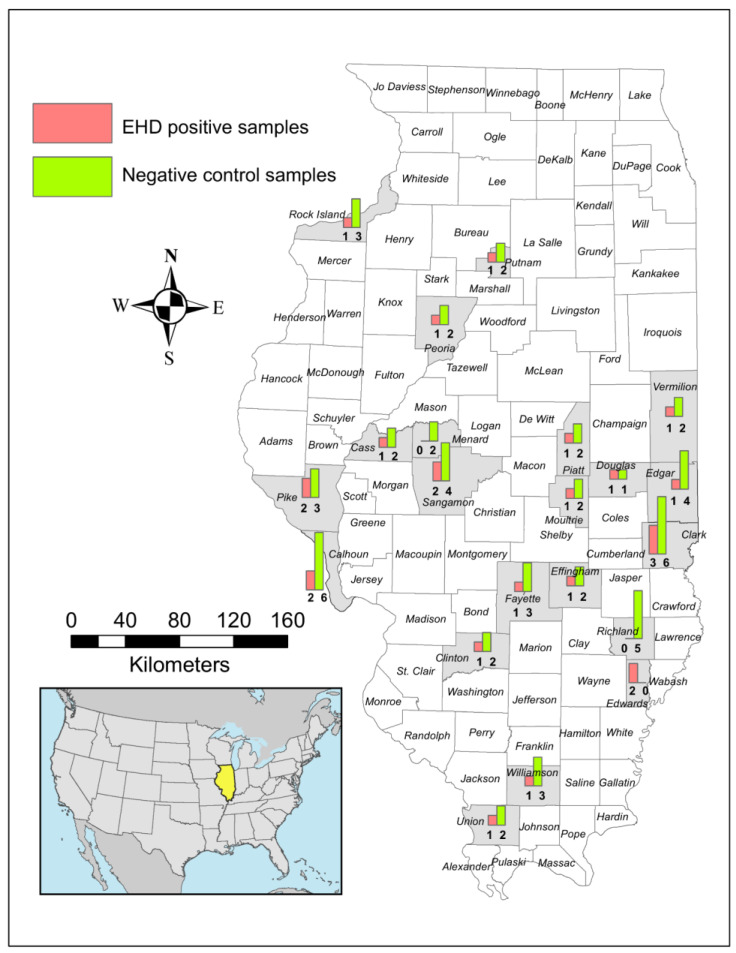
Map showing the provenance of white-tailed deer samples used in this study. Counties in Illinois from which samples were collected are shaded in gray. The number of EHD-positive deer sampled is indicated by a red bar in each county. Control sample numbers are indicated by a green bar in each county. Numbers are shown under each bar.

**Figure 2 genes-14-00426-f002:**
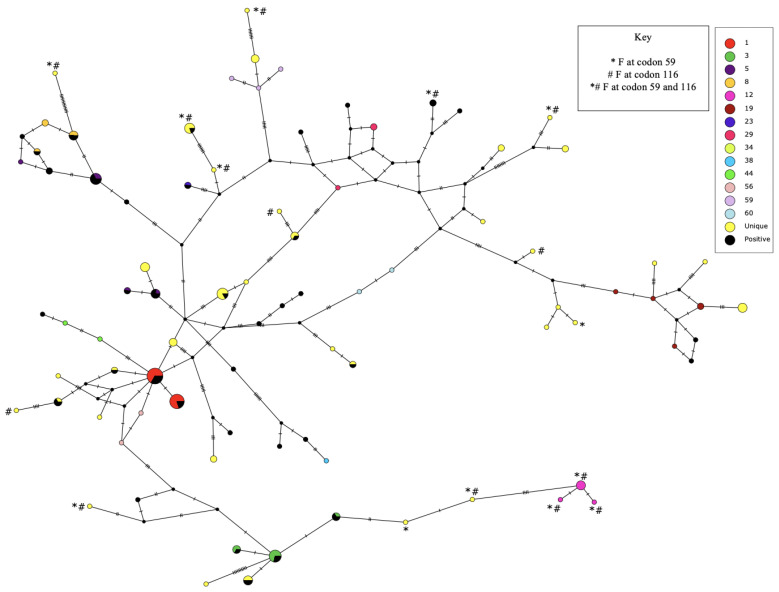
Median-joining network constructed for the 85 *TLR3* haplotypes carried by the 84 Illinois deer sequenced (2*n* = 168 chromosomes). Each distinct circle represents one unique haplotype, and the sizes of the circles are proportional to the number of chromosomes carrying the haplotype. Color shading is used to distinguish between protein variants encoded by the haplotypes; yellow coloring is used for all haplotypes that encoded a unique protein variant. Black coloring indicates the proportion of each haplotype present in deer testing positive for EHD. Hatch marks indicate the number of mutations between haplotypes. Symbols are placed next to haplotypes with SNPs that were associated with significantly reduced EHD infection: * for haplotypes encoding phenylalanine at codon 59 and # for haplotypes encoding phenylalanine at codon 116; some haplotypes carried both SNPs.

**Figure 3 genes-14-00426-f003:**
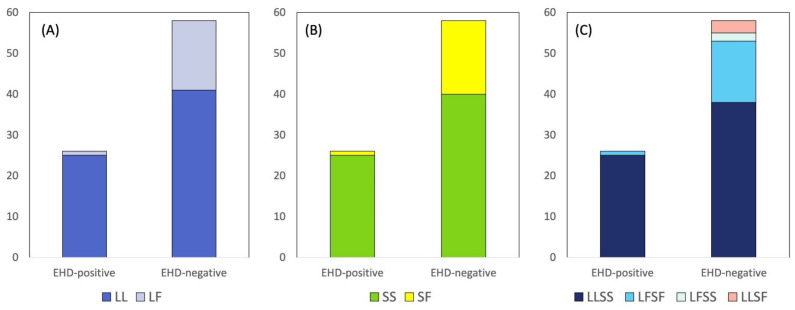
Association of SNPs in *TLR3* with epizootic hemorrhagic disease (EHD) in Illinois white-tailed deer (*n* = 84). (**A**) When both copies of the *TLR3* gene in an individual deer encoded L at codon 59, the deer were designated “LL”. When one copy encoded L and the other F, the deer were designated “LF”. In EHD-positive deer, LL was detected more frequently than LF relative to their frequencies in EHD-negative deer (*OR* = 10.16, *p* = 0.0088, Fisher’s exact test, two-tailed). (**B**) When both copies of the *TLR3* gene in an individual deer encoded S at codon 116, the deer were designated “SS”, whereas, when one copy encoded S and the other F, the deer were designated “SF”. In EHD-positive deer, SS was detected more frequently than SF relative to their frequencies in EHD-negative deer (*OR* = 11.02, *p* = 0.0048, Fisher’s exact test, two-tailed). (**C**) In the last panel, deer were designated using four letters, with the first two letters indicating the amino acids encoded by codon 59, and the last two letters indicating the amino acids encoded by codon 116. For example, deer in which both copies of *TLR3* encoded L at codon 59 and both copies encoded S at codon 116 were designated “LLSS”. The combination LLSS was detected relatively more frequently than LFSF in EHD-positive deer compared to EHD-negative deer (*OR* = 9.67, *p* = 0.0151, Fisher’s exact test, two-tailed). The absence of FF deer for codon 59 (**A**) or FF deer for codon 116 (**B**) was likely due to the low frequencies of 59F and 116F, respectively, in the population. The low frequencies of LFSS and LLSF deer (**C**) were likely due to linkage disequilibrium between the two codons (*D* = 0.083, *p* < 0.0001).

**Table 1 genes-14-00426-t001:** List of primers designed to amplify and sequence the complete coding region (in exons two through five) of *TLR3* in white-tailed deer.

Exon Targeted	Primer Name	Primer Sequence (5’ to 3’)		Uses
*Exon 2*	*OdVi-TLR3*-X2	F1:	GGGAAGGGGGAGAGTTTGTA	PCR and Sequencing
		R1:	CATATTTGAGTGTGGGGTCCC	PCR and Sequencing
*Exon 3*	*OdVi-TLR3*-X3	F3:	CCATTTTGGTGCCAAGACTA	PCR and Sequencing
		R4:	AGCCCAGACAGGAAATCAGC	PCR and Sequencing
*Exon 4*	*OdVi-TLR3*-X4	F1:	GATCAGGGAAGACCCTCTGA	PCR and Sequencing
		R1:	CCAGAGCCGAGCTAAGTTGT	PCR and Sequencing
		SF1:	CAAGCTGAGCCCCAGTCTC	Sequencing Only
		SR1:	GTCTGCTTCAGTCCATCGAA	Sequencing Only
		SF2:	CGCTCTTTTTATGGGCTTTC	Sequencing Only
		SR2:	GCCACTGAAAGGAAAAATCG	Sequencing Only
		SF3:	GCCACTGAAAGGAAAAATCG	Sequencing Only
		SR3:	TCATTTGTTAAAGTCCGCAAA	Sequencing Only
		SF4:	AGCTGACCACCAACTCTTTCA	Sequencing Only
		SR4	TTGCTTAGATCCAGAATGACCA	Sequencing Only

		F2:	CCTGGTCATTCTGGATCTAAGC	PCR and Sequencing
		R2:	ATTTCAAATGTCATAGTGTTCACC	PCR and Sequencing
		SF5:	TTTGATGAGATCCCAGTGGA	Sequencing Only
		SR5:	ACAAACCAGGCAATGCTTTC	Sequencing Only
		SF6:	CTGCTCATCCATTTTGAAGG	Sequencing Only
		SR6:	TGCTGCATATTCAAACTGCTC	Sequencing Only
		SF7:	CAGCATCAGAAGGAGCAGAA	Sequencing Only
		SR7:	ATTTCAAATGTCATAGTGTTCACC	Sequencing Only
*Exon 5*	*OdVi-TLR3*_X5	F1:	GATTTTAGAGTGTTGGGCTGTT	PCR and Sequencing
		R1:	AAGGCCTGAAATAGGGAGACA	PCR and Sequencing

## Data Availability

*TLR3* full coding sequences for white-tailed deer were deposited in GenBank under accession numbers OL744113–OL744197.

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
