# Peer review of "The Impact of Variation in the Toll-like Receptor 3 Gene on Epizootic Hemorrhagic Disease in Illinois Wild White-Tailed Deer (Odocoileus virginianus)"

_genes, 2023, doi:10.3390/genes14020426_

Round 1

Reviewer 1 Report

The paper addresses an important topic in research. The study is well-designed and describes the role of host genetics that might have on the outbreaks of EHD in deer. I find this paper well written and scientifically sound. The topic is relevant and important for public health.

Author Response

We thank the reviewer for these positive comments.

Reviewer 2 Report

This study is highly valuable for the researcher in wildlife disease, and genetic diversity of TLR3 of Illinois wild white-tailed deer. The genetic variety of TLR3 in host and EHD infectious status was analyzed using molecular methods to construct the relationship between specific allele(s) and EHD infection for predicting the protein-associated EHD infection. This knowledge is significant to determine the vulnerable host in population or geographic distribution that is crucial information for futher disease management.  

However, there is a few minor comment to correct in section of materials and methods.

line 137: "UTR" should be defined "untranslated region (UTR)" that the first appearance, then used "UTRs" (in line 140)

line 159, 163 and 166: "...an initial annealing step of 95oC for 9.45 min" the author please consideration, this step might be "an initial denaturation step" 

Author Response

This study is highly valuable for the researcher in wildlife disease, and genetic diversity of TLR3 of Illinois wild white-tailed deer. The genetic variety of TLR3 in host and EHD infectious status was analyzed using molecular methods to construct the relationship between specific allele(s) and EHD infection for predicting the protein-associated EHD infection. This knowledge is significant to determine the vulnerable host in population or geographic distribution that is crucial information for futher disease management.  

Reply: We thank the reviewer for these positive comments.

However, there is a few minor comment to correct in section of materials and methods.

line 137: "UTR" should be defined "untranslated region (UTR)" that the first appearance, then used "UTRs" (in line 140)

Reply: done

line 159, 163 and 166: "...an initial annealing step of 95oC for 9.45 min" the author please consideration, this step might be "an initial denaturation step" 

Reply: Thanks for catching this. We have corrected this.

Reviewer 3 Report

The manuscript aims to investigate the association of certain SNPs within TLR3 gene with susceptibility against epizootic hemorrhagic disease virus (EHDV) in white-tailed deer with a case-control design. Significant results were observed specifically in terms of certain non-synonymous SNPs encoding codon 59 and 116 of the TLR3 protein.  The writing of the manuscript is of great quality in terms of coherency, fluency and clarity. Study was implemented with care and expertise and results were presented clearly.

Line 159: Can you please replace the term ‘initial annealing’ with ‘initial denaturation’.  

Line 160: What do you mean by ‘three cycles’ for annealing while you specified that it is five cycles per temperature level?

Line 159-169: Can you please clarify the number of cycles per stages of PCR and levels of temperature given between these lines.

Line 199-200: Can you please briefly describe the reason for choosing a non-parametric test for the association analysis. Also between the lines 96-97, the authors have mentioned certain environmental factors such as sex and age that were recorded. I wonder how these records were incorporated through the statistical analysis process. If utilized, can you please briefly describe the statistical process for accounting for those factors? Otherwise, please remove those information given between lines 96 and 97.

Line 225: Table 2 is suggested to be given as a supplementary table and be removed from the main body of the text, as its appearance is not central to the main purpose of the study. Same applies to the Table 3.

Line 343: Please correct the referencing error ‘[Abrantes et al 2013]’.

Author Response

The manuscript aims to investigate the association of certain SNPs within TLR3 gene with susceptibility against epizootic hemorrhagic disease virus (EHDV) in white-tailed deer with a case-control design. Significant results were observed specifically in terms of certain non-synonymous SNPs encoding codon 59 and 116 of the TLR3 protein.  The writing of the manuscript is of great quality in terms of coherency, fluency and clarity. Study was implemented with care and expertise and results were presented clearly.

Reply: We thank the reviewer for these positive comments.

Line 159: Can you please replace the term ‘initial annealing’ with ‘initial denaturation’.  

Reply: Thanks for catching this. We have corrected this.

Line 160: What do you mean by ‘three cycles’ for annealing while you specified that it is five cycles per temperature level?Line 159-169: Can you please clarify the number of cycles per stages of PCR and levels of temperature given between these lines.

Reply: We have clarified the PCR steps in the text. We also added a supplementary table to show the PCR cycles involved (Table S1).

Line 199-200: Can you please briefly describe the reason for choosing a non-parametric test for the association analysis.

Reply: We have added a sentence to describe the reason for choosing a non-parametric test, in lines 198-200 that reads, “Since our data do not involve a normal distribution, and also due to the small sample counts for some SNPs, we conducted Fisher’s exact test.”

Also between the lines 96-97, the authors have mentioned certain environmental factors such as sex and age that were recorded. I wonder how these records were incorporated through the statistical analysis process. If utilized, can you please briefly describe the statistical process for accounting for those factors? Otherwise, please remove those information given between lines 96 and 97.

Reply: As suggested by the reviewer, we have removed the irrelevant information.

Line 225: Table 2 is suggested to be given as a supplementary table and be removed from the main body of the text, as its appearance is not central to the main purpose of the study. Same applies to the Table 3.

Reply: We would respectfully disagree. Table 2 lists all of the haplotype information with SNP locations, and Table 3 lists all of the protein variant information showing amino acid differences. We think that these tables contain essential information useful to the reader. Unless this is considered a required change, we would very much like to keep these tables in the main text.

Line 343: Please correct the referencing error ‘[Abrantes et al 2013]’.

Reply: done